



# Very early identification of a bimodal frictional behavior during the post-seismic phase of the 2015 $M_w$8.3 Illapel, Chile, earthquake

Cedric Twardzik[1,2], Mathilde Vergnolle[1], Anthony Sladen[1], and Louisa L. H. Tsang[1,3]

[1]Université Côte d'Azur, CNRS, Observatoire de la Côte d'Azur, IRD, Geoazur, UMR 7329, Valbonne, France
[2]now at: Institut de Physique du Globe de Strasbourg, UMR 7516, Université de Strasbourg, EOST, CNRS, Strasbourg, France
[3]now at: University of Surrey, International Study Centre, Guildford, United Kingdom

**Correspondence:** Cedric Twardzik (twardzik@unistra.fr)

**Abstract.** It is well-established that the post-seismic slip results from the combined contribution of seismic slip and aseismic slip. However, the partitioning between these two modes of slip remains unclear due to the difficulty to infer detailed and robust descriptions of how both evolve in space and time. This is particularly true just after a mainshock when both processes are expected to be the strongest. Using state-of-the-art sub-daily processing of GNSS data, along with dense catalogs of aftershocks obtained from template-matching techniques, we unravel the spatiotemporal evolution of post-seismic slip and aftershocks over the first 12 hours following the 2015 $M_w$8.3 Illapel, Chile, earthquake. We show that the very early post-seismic activity occurs over two regions with distinct behaviors. To the north, post-seismic slip appears to be purely aseismic and precedes the occurrence of late aftershocks. To the south, aftershocks are the primary cause of the post-seismic slip. We suggest that this difference in behavior could be inferred only few hours after the mainshock, and thus could contribute to a more data-driven forecasts of long-term aftershocks.

## 1 Introduction

One of the most perceptible expressions of the post-seismic activity following a large earthquake is the occurrence of aftershocks. Their temporal behavior is usually well-described by the Omori-Utsu law (Omori, 1894; Utsu et al., 1995), which states that the frequency of aftershocks decays as a power law with time after the mainshock. Still, the parameters of this law can fluctuate from one sequence to another and its physical origin is still debated (e.g., Nanjo et al., 2007; Hainzl and Marsan, 2008; Narteau et al., 2009). Regarding the spatial distribution of aftershocks, we know since the mid-1950 that it is somehow linked to the rupture area of the mainshock (Utsu and Seki, 1954). Yet, we are still unable to explain the spatial distribution for many aftershock sequences. All of that stresses our lack of understanding about what control the location and timing of aftershocks. In addition to being a fundamental question, it is also of societal interest as aftershocks can severely damage buildings already weakened by the mainshock.

With the development of seismology and geodesy over the last few decades, earthquake slip models have dramatically improved, allowing us to better investigate the relationship between the mainshock and the location and timing of aftershocks. A first-order spatial forecast can be made by stating that fewer aftershocks will occur in areas of high coseismic slip (e.g., Das





and Henry, 2003; Wetzler et al., 2018). But, the most standard approach is to forecast aftershocks where positive Coulomb
stress changes from the mainshock are experienced (e.g., Das and Scholz, 1981; Kirb et al., 2002). More recently, statistical
forecasting models have been proposed (e.g., Dasher-Cousineau et al., 2020), which tend to produce different predictions. But,
it is not clear that one approach is systematically better than another (Woessner et al., 2011).

Thanks to geodetic measurements, we know that the post-seismic activity does not only express in the form of aftershocks,
but also with aseismic slip on the fault, hereafter called afterslip (e.g., Heki and Tamura, 1997). Some studies even suggest
that it is the main driving mechanism of aftershocks (e.g., Perfettini and Avouac, 2007; Peng and Zhao, 2009; Ross et al.,
2017; Perfettini et al., 2018b, 2019). Based on that hypothesis, afterslip could be used to forecast the location and timing of
aftershocks. Compared to stress-based approaches, which rely on the modeling of both the slip distribution of the mainshock
and the Coulomb stress changes, an afterslip-based approach would rely solely on the modeling of the afterslip distribution,
potentially reducing the propagation of errors. In addition, it offers the possibility to monitor temporal changes.

A strong limitation of this approach is that the geodetic surface observations record the combined contribution of seismic
and aseismic slip on the fault. Therefore, it is crucial before all to understand how these two regimes are partitioned during the
post-seismic phase. Most of the previous studies that have investigated such issues have used daily GNSS (Global Navigation
Satellite System) position time series thus focusing on a time period starting at least from ∼12 to 24 hours after the mainshock
(e.g., Lange et al., 2014). Because of this, very little is known in the time window starting from the first minutes and up to the
first few hours after the mainshock. For instance, is afterslip the driving mechanism of aftershocks even during the very early
stage of the post-seismic phase ? Or, is the ratio between seismic and aseismic slip at this very early stage of the same order to
what is observed at longer times ?

Over the last few years, precise geodetic observations of the very early post-seismic phase with high temporal resolution
have emerged (Langbein et al., 2006; Miyazaki and Larson, 2008; Munekane, 2012; Malservisi et al., 2015; Twardzik et al.,
2019; Milliner et al., 2020) allowing us to investigate specifically these first 12 hours. At the same time, the development of
template-matching techniques using seismological data has led to the construction of more complete aftershocks catalogs (e.g.,
Shelly, 2020). These upgraded catalogs are particularly effective at detecting small earthquakes hidden in the strong seismic
noise at the beginning of the aftershock sequence. This is a critical step since the use of incomplete catalogs can lead to distorted
results (Hainzl, 2016). By combining these highly complete catalogs with geodetic observations of high temporal resolution,
we can now study in detail the relationship between aftershocks and afterslip at the very early stage of the post-seismic phase.

To this end, we investigate the first 12 hours following the September 16, 2015, $M_w 8.3$, Illapel, Chile earthquake, and for
which GNSS position time series with high temporal resolution are available over this time period (Twardzik et al., 2019)
as well as dense catalogs of aftershocks obtained from template-matching techniques (Frank et al., 2017; Huang et al., 2017).
First, we analyze the geodetic surface observations to obtain hourly images of the afterslip distribution over these first 12 hours,
and we compare that with the spatiotemporal evolution of aftershocks. Then, we estimate the seismic/aseismic slip partitioning
over this time period to better understand the mechanical behavior of the subduction interface just after the mainshock. Finally,
we discuss the potential of using very early afterslip observations for the forecast of the spatial patterns of aftershocks.





## 2 Data and Methods

### 2.1 Geodetic data

On September 16, 2015, an earthquake of magnitude 8.3 occurred near the city of Illapel in central Chile. It ruptured part of a locked segment of the south American subduction zone that is surrounded by two areas of relatively low coupling (Ruiz et al., 2016). Thanks to the Centro Sismológico Nacional (CSN) of the University of Chile, this region was heavily instrumented with GNSS stations at the time of the earthquake (Baez et al., 2018). Thus, we have access to 15 high-rate GNSS stations that are located $< 350$ km from the earthquake epicenter (Figure 1). To investigate the first 12 hours following the mainshock,

we first need to obtain sub-daily position time series. Thus, we used the 30-seconds 3-components kinematic position time series starting just 5 minutes after the origin time of the mainshock from Twardzik et al. (2019). To reduce the high frequency noise, a Kalman filter was used during the processing of the GNSS data. However it was tuned to be suitable for detecting slow processes over time scales of hours to days (Choi, 2007). Then, these time series were post-processed by applying a sidereal filter constructed in such a way that it can remove periodic noise without removing the post-seismic signal (Twardzik

et al., 2019). Finally, we disregarded the vertical component because its noise level is too large ($\sim$2 times that of the horizontal components). An example illustrating the post-processing can be found in Figure 2.

Over the first 12 hours that followed the Illapel earthquake, two large aftershocks were reported in the Global Centroid Moment Tensor catalog. The first one ($M_w 7.1$) occurred $\sim$23 minutes after the mainshock, while the second one ($M_w 6.8$) occurred $\sim$5.25 hours after it. To quantify later in the discussion the impact of these large aftershocks on the post-seismic slip,

we calculate their static offsets from the position time series. Because of the Kalman filter, these offsets are smoothed over time, and thus are not accommodated instantaneously but instead over a certain duration ($T_{\text{kalman}}$). That duration is given by the following formula: $A.\sqrt{\Delta t}.k$, where $A$ is the expected amplitude of the static offset at the stations, $\Delta t$ is the sampling interval of the time series and $k$ is the parameter of the Kalman filter ($3.0 \times 10^{-7}$km/$\sqrt{s}$). We estimate $A$ by computing the expected offsets using Okada's formulas (Okada, 1985) inside an homogeneous half-space ($\mu = 39$ GPa for the rigidity and

$\nu = 0.25$ for the Poisson's ratio) and using the nodal plane with the shallower dipping angle as well as the hypocenter location reported by the GCMT catalog. The expected width and length of the fault plane is determined using the empirical relationship for subduction-interface events obtained by Thingbaijam et al. (2017). Using the estimated $T_{\text{kalman}}$, we extract the static offsets of these two aftershocks from the position time series by choosing a pre-earthquake position at $t = $ origin time $- T_{\text{kalman}}$ and a post-earthquake position at $t = $ origin time $+ T_{\text{kalman}}$. We center the offset on the origin time of each earthquake to account for

the fact that the Kalman filter is applied both forward and backward. Tables **??** and **??** summarize the computed offsets at each station for both aftershocks, and an example of a corrected time series is presented in Figure 2.

The kinematic position time series, even after applying a sidereal filter, remain relatively noisy, with the standard deviation calculated from 6 days of data prior to the mainshock ranging from 2.45 to 3.98 mm. Thus, we choose to favor noise reduction over the rate of positioning by smoothing the time series using a 1-hour sliding window. The cumulative surface displacements

are calculated at every hour since the mainshock origin time by computing the average positions over a 1-hour time window centered on the time of interest. The errors associated with these new observations are set to the standard deviation of the time





series measured prior to the mainshock. Figure 2 illustrates for one station the different steps that we perform to obtain the hourly position time series (see Supplementary Material S1 for all the stations). Because of strong spurious signals that are not of tectonic origin, we disregard the first two hours for station LVIL and hours 2 and 3 for station OVLL.

## 95 2.2 Inversion of the very early afterslip

Using the observations described above, we attempt to obtain the hourly spatial distribution of afterslip on a planar fault that is 600 km long along strike and 300 km long along dip (Figure 1). Our assumed fault geometry has a strike of 3°N and is adjusted such that its upper edge coincides with the trench. The fault dip is chosen to be 17°, which corresponds to the average dip of the slab at this location calculated from the Slab1.0 model from Hayes et al. (2012). Then, the fault is divided into 100 450 sub-faults, 30 along strike and 15 along dip, with the slip amplitude and rake angle evaluated at their centers. We use the approach by Zhu and Rivera (2002) to compute the static response of each sub-fault in a tabular media obtained from the CRUST1.0 database (Laske et al., 2013). For a given source model, the surface displacements at each receiver are computed by summing the contribution of each sub-fault.

We search for the spatial distribution of slip amplitude and rake angle independently for each time step. The spatial distri-
bution of the source parameters is obtained using an optimization procedure similar to that of Pianatesi et al. (2007). We use a heat-bath simulated annealing algorithm, an evolution of the Metropolis-Hastings that lowers the rejection rate by computing the relative probabilities from a set of trial models before a random move is made (Sen and Stoffa, 2013). To measure the difference between the observed ($o$) and calculated ($c$) surface displacements, we use the cost function of Pianatesi et al. (2007) to which we add a smoothing constraint:

$$\varepsilon = \frac{1}{N} \frac{\sum\limits_{i=1}^{N} \left( (o_i - c_i)/e_i \right)^2}{\sum\limits_{i=1}^{N} o_i^2} + \omega \Lambda \qquad (1)$$

where $N$ is the number of observations (2 horizontal components $\times$ number of receivers), $e$ is the error associated with the observation, $\Lambda$ is the Laplacian of the slip distribution and $\omega$ is the weight given to the Laplacian (0.1 in this study - see Supplementary Material S2 for a discussion on how the value is chosen).

Similarly to Pianatesi et al. (2007), we keep track of the model after each iteration. For each time step, we run the algorithm 115 100 times, each time with a different random seed, which leads to a slightly different outcome after each run. Thus, we end up for each time step with an ensemble of 40,000 models along with their misfit values. We use this ensemble of models to build an average one along with its standard deviation. Both are computed by weighting the models by the inverse of their misfit, thus giving more weight to best fitting models. Hereafter, we set to zero the sub-faults that have a mean slip amplitude that is smaller than its standard deviation.



## 3  Spatiotemporal evolution of the very early post-seismic slip

Following the procedure described above, and using the time series recording both seismic and aseismic deformation, we obtain hourly post-seismic slip distribution models. This allows us to investigate its spatiotemporal evolution over the first 12 hours (Figure 3 - The fit to the observations is shown in Supplementary Material S3). Our results show distinct patches that are not moving in space but grow in amplitude over time, and that can be identified 1 to 3 hours after the mainshock. The first one to develop is located near the epicenter of the mainshock, off-shore of Canela Baja (station CNBA). This is also where the 2 largest aftershocks are located (yellow circles on Figure 1). This patch seems to nucleate at the southern edge of the co-seismic rupture area inferred by Melgar et al. (2016). The latter is chosen because it is mostly based on data recording the co-seismic phase only (i.e., seismological data, high-rate GNSS data, tsunami data), limiting the contamination from very early afterslip by the InSAR data. After 12 hours, we find that the geodetic moment of this patch is $3.7 \times 10^{19}$ Nm ($M_{w_{eq}}$7.0). The second to develop is located to the north of the co-seismic rupture area, off-shore of Fray Jorge Park (station PFRJ). Although some slip is seen in this area during the first 2 hours (<0.1 m), this patch starts to grow noticeably in amplitude after 3 hours, and ends up with a geodetic moment of $1.5 \times 10^{19}$ Nm after 12 hours ($M_{w_{eq}}$6.7). In between these two patches, and down-dip of the co-seismic rupture area, a connection starts to robustly build-up ∼7 hours after the mainshock. After 12 hours, we find that the model has a geodetic moment of $1.0 \times 10^{20}$ Nm ($M_{w_{eq}}$7.3), and it shows a rather continuous region of slip that surrounds the area of co-seismic rupture. When compared with models of post-seismic slip over longer time scales (from 1 day up to 2 months after the mainshock; see Figure **??**), we find that the bimodal slip distribution persists over time. Thus, it appears that following the Illapel earthquake, the post-seismic slip patches remain at a steady location throughout the first 2 months. Thus, in this specific case, the information about the early-stage of the post-seismic phase could have been extrapolated to infer the longer-term slip pattern.

When we look more closely, we see that some of the post-seismic slip might have penetrated inside the co-seismic rupture area (Figure 3). There have been such observations at the very early stage of the post-seismic phase in particular following the 2008 Tokachi-Oki, Japan, earthquake (Miyazaki and Larson, 2008) and the 2016 Pedernales, Ecuador, earthquake (Tsang et al., 2019). Under the standard rate-and-state framework usually used to explain post-seismic slip, this should be precluded because co-seismic slip is associated with slip-weakening frictional conditions while post-seismic slip is rather associated with slip-strengthening frictional conditions (e.g., Marone et al., 1991). However, there are many lines of evidences indicating that the frictional properties of a fault might not be steady over time, because of a redistribution of stresses in the surrounding medium after a large earthquake (Helmstetter and Shaw, 2009), heterogeneities in the mineral composition of fault gouges (Colletini et al., 2011) or complexities in the fault geometry (Romanet et al., 2018). When we compare our post-seismic slip model with other co-seismic slip models, we see that our slip model lies at the edges of most of the co-seismic slip models. However, depending on which model we choose for the later, the amount of overlap varies significantly (see Supplementary Material S4). Given the variability between the different co-seismic models and the level of uncertainty in our own post-seismic slip models, we find it difficult to reach a definitive conclusion on the matter. This is in line with Barnhart et al. (2016) who discuss this question by looking at 26 days after the mainshock and using a similar dataset to ours (i.e., 9 of the 15 GNSS





stations plus InSAR data). They also find some overlap between co-seismic and post-seismic slip but the authors conclude that
this result is not robust enough to be interpreted.

Finally, we also identify in our models a region where significant slip occurs and that is located south-east of Salamanca (station SMLC). This region is further away from the co-seismic rupture area and is disconnected from the main regions of post-seismic slip. After some tests (see Supplementary Material S5), we conclude that this patch is an unreliable feature and is therefore disregarded in the discussion section. When this region is not taken into account, the final geodetic moment decreases
to $8.3 \times 10^{19}$ Nm ($M_{w_{eq}}$ 7.2).

## 4  Discussion

### 4.1  Relationship between very early aftershocks and very early post-seismic slip

One of the first points we aim to assess is the relationship in space and time between aftershocks and post-seismic slip right after the mainshock, here during the first 12 hours. For that, we compare the spatiotemporal evolution of slip and aftershocks
using the catalog compiled by Huang et al. (2017) (see Figure **??**). Although it is possible that some of these aftershocks are not on the subduction interface, based on the focal mechanisms obtained by Carrasco et al. (2019) we can make that assumptions at least for regions where we observe afterslip. With our comparison, we find that both show large similarities. Over the first 2 hours, most of the seismic activity occurs south of the rupture area, just as does the post-seismic slip. To the north, very little activity is seen during these first 2 hours but it then progressively intensifies just like the post-seismic slip. After the first 12
hours of the post-seismic phase, we see a pattern where post-seismic slip and aftershocks strongly overlap, both surrounding the co-seismic rupture area. The same observation can be made when we use the independently obtained aftershocks catalog by Frank et al. (2017) (see Supplementary Material S6).

The strong similarity between these two spatiotemporal evolutions could be due to the cumulative contribution of aftershocks that are not removed from the position time series, thus contributing to the estimated fault slip. For instance, this is what has
been proposed to explain the pre-seismic slip pattern prior to the 2014 Iquique, Chile, earthquake (Schurr et al., 2014). This naturally leads to the question of the partitioning between seismic and aseismic slip during the very early post-seismic phase. To investigate this question, we compare the geodetic moment from the post-seismic slip models with the seismic moment released by aftershocks. To estimate the latter, we use the GCMT catalog for the 2 largest earthquakes and the catalog from the Centro Sismológica Nacional (CSN) of the University of Chile for the smaller earthquakes, restricting to those with available
moment magnitude estimates. We thus consider 38 aftershocks with magnitudes ranging from 4.5 and up to 7.1 that give a seismic moment of $9.5 \times 10^{19}$ Nm. Hence, about 95% of the post-seismic slip can be imputed to seismic slip rather than aseismic slip.

However, there is a very large discrepancy between the patches: ~95% of the seismic moment released by aftershocks happens inside the southern patch, with the 2 largest events accounting for ~85% of the total moment. Hence, to better isolate
aseismic slip, we use corrected position time series, of which the contribution of these two largest aftershocks is removed, to obtain hourly images of the post-seismic slip. The results can be seen in Figure **??**. Our models show that post-seismic slip





is now only observed north of the co-seismic rupture area and that the patch to the south has completely vanished. Therefore, post-seismic slip south of the mainshock rupture area is likely the result of seismic slip only. On the contrary, post-seismic slip to the north is likely aseismic as the contribution of aftershocks ($1.2{\times}10^{17}$ Nm) is small compared to the geodetic moment

of this patch ($7.3{\times}10^{18}$ Nm). Some clues that the southern patch results primarily from seismic slip could have also been inferred by looking at the temporal evolution between post-seismic slip and seismic moment from aftershocks in this region (see Supplementary Materials S7). Indeed, we find that in the southern patch, the cumulative post-seismic slip follows the cumulative seismic moment released by aftershocks, which suggests a strong link between the imaged slip and slip from aftershocks.

As a next step, we analyze the temporal evolution between the amount of afterslip and the number of aftershocks in the northern patch (Figure **??**). First, we find that the cumulative afterslip exhibits a logarithmic trend. Similarly to other studies (e.g., Tsang et al., 2019; Milliner et al., 2020), we do not observe the acceleration phase of afterslip predicted by Perfettini and Ampuero (2008). This suggests that for this earthquake also, the steady-state is reached very quickly after the mainshock ($<$ 1 hour). When we make the comparison with the cumulative number of aftershocks, we find that the two curves do not follow

the same trend. Instead the cumulative afterslip relates with the log of the cumulative number of aftershocks. This seems at odds with observations made on the early times of the post-seismic phase by Tsang et al. (2019) and Milliner et al. (2020). As these two studies do not specifically focus on the first 12 hours but rather on the trend over a couple of days, it would be very interesting to investigating in detail what happens during the very early stage of the post-seismic phase for these two examples.

Based on theoretical arguments, Perfettini and Avouac (2004) suggest that if afterslip is driving the generation of aftershocks,

we should expect a similar time evolution between the cumulative number of aftershocks and the cumulative afterslip. Their analysis assumes that the deformation from aftershocks should be small compared to that of afterslip and that steady-state is reached, and both criteria seem to be met here. The fact that we do not observe such relationship is not incompatible with the fact that afterslip drives aftershocks even just after the mainshock. The way aftershocks respond to stress changes may be more complex, especially when considering the full rate-and-state law (e.g., Dieterich, 1994; Kaneko and Lapusta, 2008; Helmstetter

and Shaw, 2009). It is also possible that very early aftershocks are triggered by the mainshock static stress changes and that it is only later that afterslip starts to drive aftershocks (Perfettini et al., 2018a). Other processes could also be candidates as the driving mechanism for aftershocks generation (e.g., fluid flow, Miller, 2020). We also cannot rule out the fact that the earthquake catalog might still be incomplete thus missing aftershocks at the very early stage of the post-seismic phase.

To summarize, we find here that post-seismic slip can arise from distinct frictional properties on the fault. To the south, slip

is essentially seismic, which can be usually associated to a velocity-weakening regime under the rate-and-state framework. Instead, to the north, slip is almost purely aseismic, which is more inline with a velocity-strengthening regime. We can hypothesize regarding the causes of that bimodal behavior. For instance, Poli et al. (2017) suggest that fracture zones enclose the mainshock rupture area favoring fluids circulation at these places. Differences in the pore-fluid pressure between the north and the south could be why we observe distinct behaviors. We also find that the southern patch is associated with a high coupling

of the plate interface while the one to the north is at a transition zone between high and low coupling (Métois et al., 2013). This could explain why one region favors seismic slip (south) while the other favors aseismic slip (north). Still, the fact that





we can very early-on suggest that the fault is divided into regions of distinct frictional properties can prove to be useful when forecasting aftershocks using physics-based models.

## 4.2 Potential to use very early post-seismic to forecast aftershocks location

Most of the physics-based models of aftershocks sequences used for operational forecasting do not include information about post-seismic slip, although the study by Cattania et al. (2015) shows that it could have a positive impact on the forecast.

In the case of the Illapel earthquake, we show that the surface observations carry enough information about the spatiotemporal evolution of post-seismic slip very early in the post-seismic phase. We also see that the slip pattern inferred at this very early stage does not evolve much over time. This is not an isolated example as Tsang et al. (2019) made a similar inference for
the 2016 Pedernales, Ecuador, earthquake, with the slip patches imaged over the first 72 hours persisting after 30 days. Milliner et al. (2020) provides another example of such behavior following the 2016 Kumamoto, Japan, earthquake. Thus, post-seismic slip from the first hours after the mainshock can help characterize longer-lasting post-seismic slip. Finally, we also show that differences in frictional behavior can be revealed from the very first stage of the post-seismic phase. Thus, imaging the very early post-seismic slip with little time latency could provide valuable additional information to help the operational forecasting
of aftershocks with physics-based models.

Getting images of the slip distribution is not the main challenge as the problem is linear and computationally rather fast. The challenge lies more in the ability to obtain rapidly GNSS position time series with a low-enough noise level. Thanks to the growing number of GNSS networks worldwide, along with the improvement of the processing techniques, we find an increasing number of examples showing that it is possible to accurately monitor in near real-time various geophysical processes,
whether these processes involve seismic slip and/or aseismic slip. For instance, such monitoring is done for the rapid study of earthquakes (e.g., Murray et al., 2018; Melgar et al., 2019), or to monitor volcanic activities (e.g., Neal et al., 2019), but with an accuracy of only a few centimeters.

To the best of our knowledge, the Nevada Geodetic Laboratory (NGL) provides since 2019 an open-access to 5-minute position time series with a 1-hour latency (so-called ultra-rapid solutions), and for a significant number of GNSS stations
worldwide (see Blewitt et al. (2018) for details about the data available at the NGL). However, the short-latency at which these solutions are made available is associated with an increase of the noise level compared to the solution used in this study. Therefore, we first have to assess if the noise level of these observations is low-enough to infer information about the very early post-seismic slip.

Thus, we compare the noise level of the position time series from the two stations of our study which are also available
from the NGL, CERN and SANT. We choose the period of July 2019 as there are no significant earthquakes reported nearby in the GCMT catalog. The average standard deviation of the ultra-rapid NGL time series is $0.95\,\mathrm{cm}$ and $1.31\,\mathrm{cm}$ for the East and North components, respectively, which is higher than for the post-processed time series used in this study ($0.26\,\mathrm{cm}$ and $0.30\,\mathrm{cm}$ for the East and North components, respectively). But, we note that 5 out of 15 stations that we use show surface displacements that are larger than $1.31\,\mathrm{cm}$ after just 1 hour. Thus, given the 1-hour latency before the NGL ultra-rapid position
time series are made available, it would have been possible to obtain an image of the first hour of the post-seismic slip $\sim 2$





hours after the occurrence of the mainshock. This preliminary conclusion is promising in the prospect of including information about very early post-seismic slip in the operational forecasting of aftershock locations.

## 5 Conclusions

Over the first 12 hours following the 2015 $M_w$8.3 Illapel earthquake, we find that very early post-seismic slip develops essentially over two regions located on the edges of the co-seismic rupture area. When compared to the spatiotemporal of the very early aftershocks, we find a good spatial correlation. However, the underlying physics driving the slip in these two regions are different. To the south, we show that post-seismic slip is purely seismic and caused by the occurrence of aftershocks. Once we account for the two largest ones, slip in this region vanishes. To the north, we find that post-seismic slip is almost purely aseismic. Our findings show that in this region, afterslip does not exhibit the acceleration phase predicted from theory suggesting a rather fast transition from co-seismic to steady-state post-seismic ($<$ 1 hour). As to whether or not very early afterslip is the driving mechanism of very early aftershocks, we cannot provide a clear answer. The unusual relationship between the time evolution of afterslip and aftershocks that we obtain is not incompatible with the fact that afterslip drives aftershocks, but other hypotheses can also be proposed (e.g., delay in the role played by afterslip, influence of fluid flow, etc.). These observations of the very early post-seismic phase, at high temporal resolution, could thus be crucial to discriminate between these competing processes.

Our additional finding is that the slip patterns that we observe after 12 hours persists over the first 2 months. When that is the case, information about very-early post-seismic slip can help to characterize longer-lasting post-seismic slip, which can prove to be useful to include for the forecast of aftershocks locations. In this perspective, we suggest that an image of the first hour of post-seismic could be obtained within ∼2 hours after the mainshock origin time when using ultra-rapid position time series such as those computed at the NGL. Thus, future studies should test the capability to image very early post-seismic slip in near-real-time and investigate what this new piece of information would bring for the forecasting of the spatial distribution of aftershocks.

*Code and data availability.* The 30-seconds kinematic position time series used in this study can be found here: https://github.com/cedrictwardz/2015_IllapelEarthquake_PositionTimeSeries.git. A Python code used for building and applying the sidereal filter to the raw position time series can be found here: https://github.com/cedrictwardz/SiderealFilter.git.

*Author contributions.* CT processed the GNSS data, performed the post-seismic slip inversion and constructed the manuscript with inputs from all co-authors.

*Competing interests.* The authors declare that they have no conflict of interest.



*Acknowledgements.* The continuous GNSS stations are part of the Chilean GNSS network maintained by the Centro Sismológico Nacional of

the University of Chile (Baez et al., 2018). We thank Christophe Vigny for facilitating the access to the GNSS data. This work was conducted within the ANR JCJC E-POST (ANR 14-CE03-0002-01JCJC E-POST). CT was supported by the Centre National d'Études Spatiales, the ANR JCJC E-POST and the European Research Council (ERC, under the european union's horizon 2020 research and innovation programme under grant No. 805226).





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


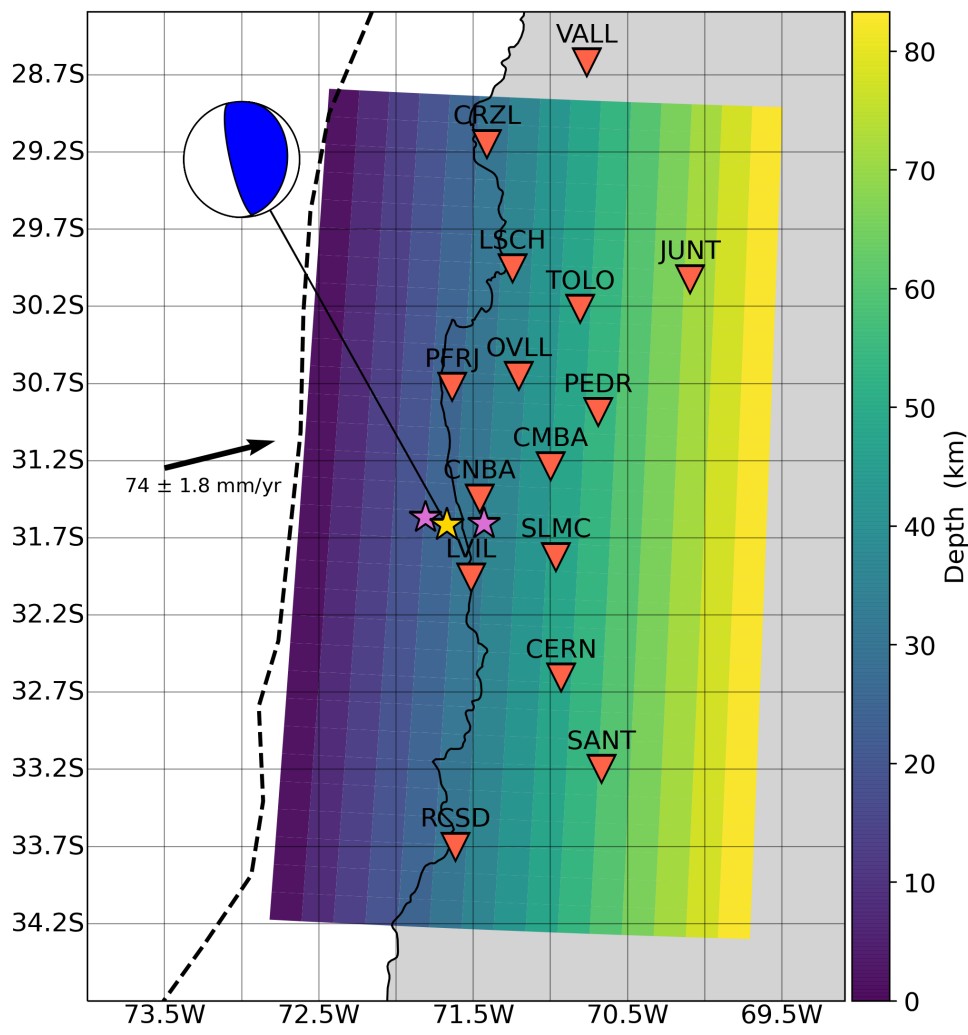

**Figure 1.** Setup used for the post-seismic slip inversions presented in this study. The fault plane (3°N of strike and 17° dipping angle) is color-coded according to depth. The red triangles are the GNSS stations. The yellow star shows the epicenter of the 2015, M$_w$8.3, Illapel, Chile earthquake and it is connected to its focal mechanism retrieved from the United States Geological Survey. The yellow circles are the largest aftershocks reported by the Global Centroid Moment Tensor (GCMT) catalog (M$_w$6.8 and M$_w$7.1 west and east of the yellow star, respectively). The dashed line shows the location of the plate boundary from Bird (2003). Finally, the black arrow shows the plate motion of the Nazca plate with respect to a fixed South America plate (DeMets et al., 2012).



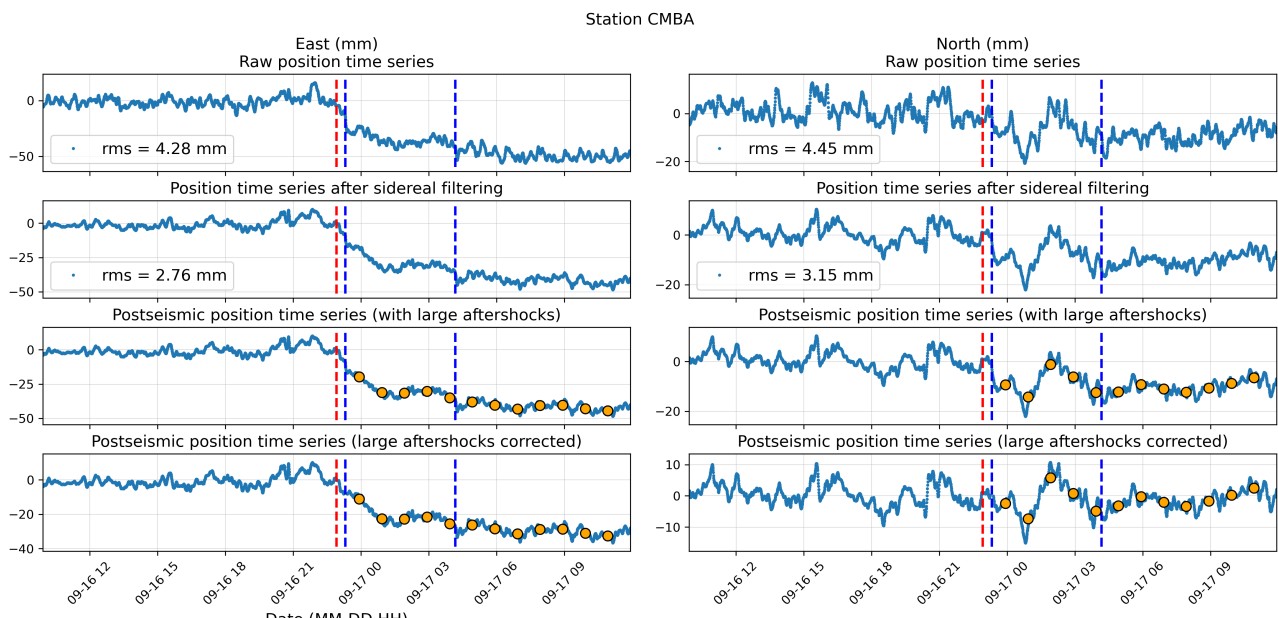

**Figure 2.** Processing steps performed to get the time series used in this study. The first row shows the raw position time series. The mean and a linear trend have been removed using the 6 days prior to the mainshock origin time (red dashed line). The standard deviation is also calculated from the 6 days prior to the mainshock. The co-seismic static offset of the mainshock is removed from the time series. The second row shows the position time series after applying a sidereal filter which is constructed as proposed by Twardzik et al. (2019). This allows to reduce the noise level by ∼35 % across all stations. The third row shows the position time series that we use to obtain the spatiotemporal evolution of the post-seismic slip over the first 12 hours (orange dots). Each dot is the average position using a 1-hour time window centered on the time of interest and spanning 30 minutes on either side. The fourth row shows the position time series with the estimated co-seismic static offsets of the 2 largest aftershocks removed. The blue dashed lines show the 2 largest aftershocks in the GCMT catalog. Supplementary Material S1 shows a similar figure for all the stations.



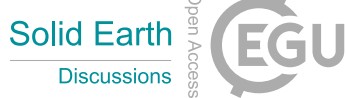

**Figure 3.** Hourly post-seismic slip over the first 12 hours after the mainshock. Each model is obtained after averaging 40,000 models, each weighted by the inverse of its misfit value. Sub-faults with slip amplitude lower than the standard deviation are set to 0. The light blue area shows the co-seismic slip region obtained by Melgar et al. (2016). The blue stars show the 2 largest aftershocks in the GCMT catalog at their time of occurrence.



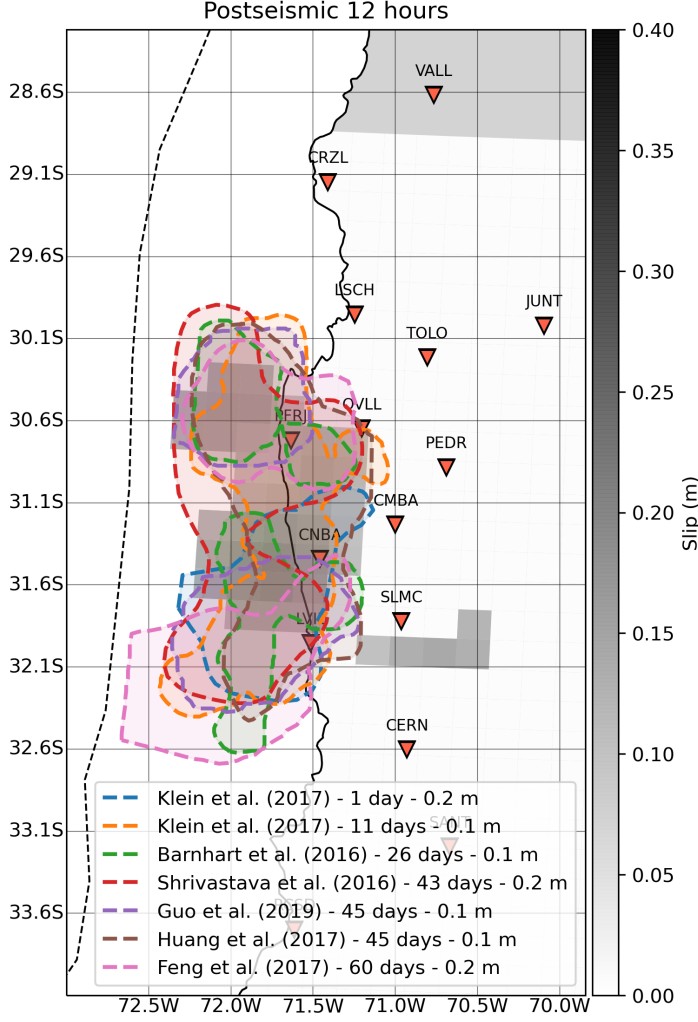

**Figure 4.** Comparison between post-seismic slip after 12 hours (this study) and post-seismic slip after 1 day and up to several weeks after the mainshock (dashed lines). Klein et al. (2017) use GNSS data and look at the shorter time period (1 day and 11 days). Barnhart et al. (2016) use both GNSS and InSAR data to model post-seismic slip after 26 days. Shrivastava et al. (2016) use GNSS data to investigate post-seismic slip after 43 days. Guo et al. (2019) also use GNSS data to image post-seismic slip after 1.5 months. The post-seismic slip distribution taken from Huang et al. (2017), using GNSS and InSAR data, is also after 1.5 months. Finally, the model obtained by Feng et al. (2017) uses geodetic data to look at post-seismic slip after 2 months. Numbers inline the references is the slip isoline amount. Symbols are the same as for Figure 1.





**Figure 5.** This is the same as Figure 3. In addition, the cumulative aftershocks from the catalog compiled by Huang et al. (2017) are shown with black dots.







**Figure 6.** This is the same as Figure 3 except that these hourly images have been obtained after removing the 2 largest aftershocks from the position time series.





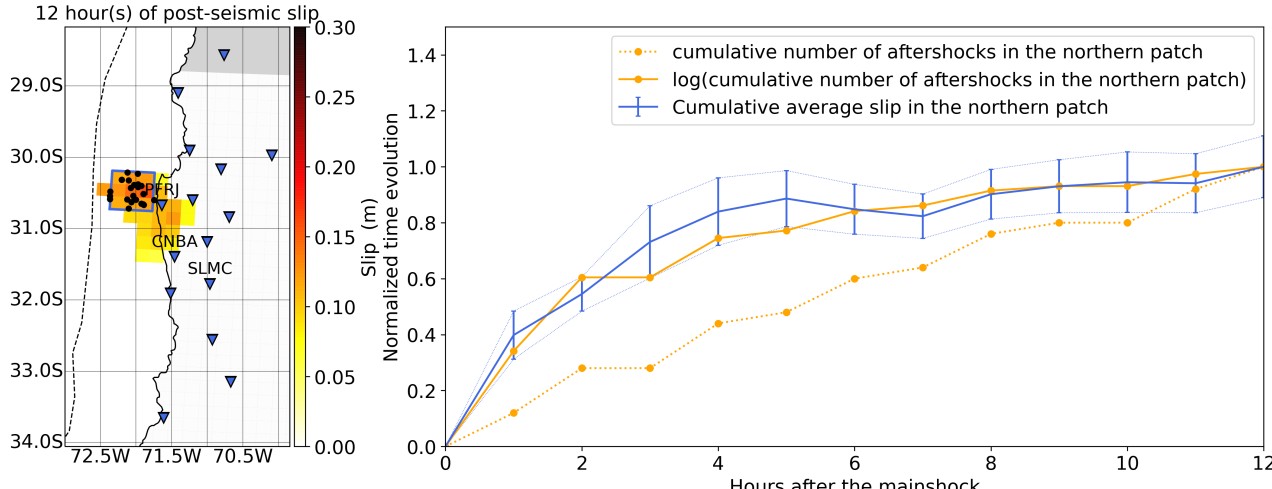

**Figure 7.** (left) Post-seismic slip distribution after 12 hours and obtained using the position time series with the 2 largest aftershocks removed. We show with the blue rectangle the identified northern patch that is slipping almost purely aseismically. The black dots show the aftershocks inside this region from the catalog of Huang et al. (2017). See Figure 1 for details on the symbols. (right) Time evolution of the cumulative afterslip (blue line) along with the time evolution of the cumulative number of aftershocks (orange dotted line). We show with the orange solid line that cumulative afterslip relates non-linearly with the log of the cumulative number of aftershocks. All curves are obtained from the inside of the blue rectangle shown on the left.





**Table 1.** Static offsets inferred from the 30-seconds position time series for the $M_w 7.1$ aftershock. See Figure 1 for the location of each station. We also show the theoretical duration over which the offset is smoothed because of the Kalman filter.

| Station name | North (mm) | East (mm) | $T_{\text{kalman}}$ (s) |
|---|---|---|---|
| LSCH | $0.06 \pm 3.09$ | $0.09 \pm 2.98$ | 30.0 |
| JUNT | $0.01 \pm 3.22$ | $-0.35 \pm 3.34$ | 30.0 |
| CRZL | $0.08 \pm 2.92$ | $-0.10 \pm 2.65$ | 30.0 |
| RCSD | $-0.32 \pm 3.21$ | $0.36 \pm 2.78$ | 30.0 |
| CERN | $3.38 \pm 3.50$ | $0.51 \pm 3.43$ | 90.0 |
| SANT | $0.33 \pm 2.85$ | $0.01 \pm 2.93$ | 30.0 |
| PEDR | $1.00 \pm 3.48$ | $-4.53 \pm 2.72$ | 120.0 |
| PFRJ | $0.34 \pm 3.00$ | $0.24 \pm 2.63$ | 30.0 |
| LVIL | $23.71 \pm 3.98$ | $-22.02 \pm 3.06$ | 510.0 |
| TOLO | $0.52 \pm 3.10$ | $-0.88 \pm 2.45$ | 30.0 |
| CMBA | $-6.95 \pm 3.15$ | $-8.67 \pm 2.76$ | 240.0 |
| OVLL | $0.63 \pm 3.45$ | $0.30 \pm 3.38$ | 60.0 |
| SLMC | $8.28 \pm 3.29$ | $-24.55 \pm 3.03$ | 450.0 |
| CNBA | $-12.79 \pm 3.05$ | $-18.23 \pm 2.75$ | 360.0 |
| VALL | $0.02 \pm 3.16$ | $0.06 \pm 3.06$ | 30.0 |





**Table 2.** Static offsets inferred from the 30-seconds position time series for the $M_w 6.8$ aftershock. See Figure 1 for the location of each station. We also show the theoretical duration over which the offset is smoothed because of the Kalman filter.

| Station name | North (mm) | East (mm) | $T_{\text{kalman}}$ (s) |
|---|---|---|---|
| LSCH | $0.02 \pm 3.09$ | $-0.09 \pm 2.98$ | 30.0 |
| JUNT | $0.03 \pm 3.22$ | $-0.04 \pm 3.34$ | 30.0 |
| CRZL | $0.16 \pm 2.92$ | $0.40 \pm 2.65$ | 30.0 |
| RCSD | $-0.06 \pm 3.21$ | $0.34 \pm 2.78$ | 30.0 |
| CERN | $-0.22 \pm 3.50$ | $-0.17 \pm 3.43$ | 30.0 |
| SANT | $0.16 \pm 2.85$ | $-0.07 \pm 2.93$ | 30.0 |
| PEDR | $0.02 \pm 3.48$ | $-0.31 \pm 2.72$ | 30.0 |
| PFRJ | $0.01 \pm 3.00$ | $0.30 \pm 2.63$ | 30.0 |
| LVIL | $-0.24 \pm 3.98$ | $-1.99 \pm 3.06$ | 90.0 |
| TOLO | $-0.04 \pm 3.10$ | $0.19 \pm 2.45$ | 30.0 |
| CMBA | $-2.03 \pm 3.15$ | $-3.17 \pm 2.76$ | 90.0 |
| OVLL | $-0.32 \pm 3.45$ | $-0.10 \pm 3.38$ | 30.0 |
| SLMC | $-0.77 \pm 3.29$ | $-1.27 \pm 3.03$ | 120.0 |
| CNBA | $-5.67 \pm 3.05$ | $-6.93 \pm 2.75$ | 150.0 |
| VALL | $0.16 \pm 3.16$ | $-0.41 \pm 3.06$ | 30.0 |