# Peer review of "Very early identification of a bimodal frictional behavior during the post-seismic phase of the 2015 $M_w 8.3$ Illapel, Chile, earthquake"

_Solid Earth, 2021_

## Author Response (AR1)

Dear editor, reviewers, and community reviewers,

First of all, we would like to thank you for reviewing our manuscript se-2021-6 untitled « Very early identification of a bimodal frictional behavior during thepost-seismic phase of the 2015 Mw8.3 Illapel, Chile, earthquake » that we have submitted to EGU Solid Earth. We have carefully considered all the points that were raised and have attempted to address them in this revised version of the manuscript. Below, you will find a point-by-point response to all the comments. The questions/comments are in blue while our response is written in black. Note that the line numbers are the one from the manuscript with tracked-changes.

**Community Comment #1 by Sylvain Barbot:**

- Abstract lines 1-3: a lot of "slip" in just a few sentences.

We have rewritten the beginning of the abstract so that we avoid such repetitions of the word slip.

- Line 34: unclear what "potentially reducing the propagation of errors" means.

What we meant to say was that the approach of forecasting aftershocks using Coulomb stress changes relies on the modeling of the slip distribution as well as the modeling of the Coulomb stress changes. Thus, for the modeling part alone, there are potentially two sources of errors. On the contrary, forecasting aftershocks solely based on afterslip implies only one source of error for the modeling part. We have rephrased that sentence to make that point clearer (line 34-36 of the new manuscript).

- Line 46-48: detection of early aftershocks after the Gorkha earthquake was discussed in the study

Thank you for drawing our attention to that study. We have included it in the article to answer that comment, which has also been raised by reviewer Bernd Schurr (line 52-54 of the revised manuscript).

- Line 64: 15 GNSS stations within 350 km does not sound like much. Discussion of resolution and sensitivity is in order. & Lines 104-113: Not sure why a Monte Carlo sampling method is used here as the problem is entirely linear and can be solved by least squares with Laplacian regularization. It would be useful to document the resolution of the inverse problem or to characterize it with a checkerboard test.

We have added a resolution analysis to the Supplementary Material S5. More specifically, we show a map of the resolution in Figure S5.1 (i.e., diagonal elements of the resolution matrix -- Tarantola and Valette 1982). We also show on Figure S5.1 a map of the restitution for the two major patches discussed in the main text to show the potential smearing effect when slip is imaged in these areas.

- Line 137: I can't recall an example of the opposite. Do we have examples of afterslip distributions that are firmly not time/space separable ?

After doing some literature review, we have to agree with the reviewer that it is difficult to find examples showing afterslip migration. So far, we could only find one study suggesting hints for afterslip migration : Jiang et al. (2021). Even in that study, the migration is not so much on the causative fault plane, but more migration of afterslip to adjacent faults. Therefore, it is true that the most common observation is that afterslip is stable over time. We have modified the text to reflect that (line 157-161 of the revised manuscript).

- Line 143-144: Note the work of Salman, R., Hill, E.M., Feng, L., Lindsey, E.O., Mele Veedu, D., Barbot, S., Banerjee, P., Hermawan, I. and Natawidjaja, D.H., 2017. Piecemeal rupture of the Mentawai Patch, Sumatra: the 2008 mw 7.2 North Pagai earthquake sequence. Journal of Geophysical Research: Solid Earth, 122(11), pp.9404-9419.

We have added that reference in the main text as a potential mechanism for explaining an overlap between co-seismic slip and afterslip (line 172 of the new manuscript). However, we want to emphasize that, as stated in the text, our conclusion is that the resolution of our afterslip model along with the variability of coseismic models does not provide enough evidences to conclude with certainty if this is the case or not.

- Lines 156-158: Is it possible that this deep slip patch may in fact represent strain on crustal faults above the megathrust ?

This is an interesting point that is raised here. However, we do not think that this is the case. We have looked at the contribution of this patch alone on the surface displacements (Supplementary Materials S5). When looking at that, we find that this patch do not generate a significant signal at the surface. In addition, we also provide in Supplementary Material S5 two qualitative arguments that in our opinion reflect the fact that this patch is unreliable : (1) its spatio-temporal evolution is not very stable compared to the main areas of afterslip (see Figure 3 in the main text) and (2) it almost completely disappears when we use the position time series corrected from the two large aftershocks (see Figure 7 in the main text). This is why we have reached the conclusion that this patch is rather uncertain and more likely an artefact from the inversion.

- Line 181: Wasn't the geodetic moment 8E19 Nm after the deep patch is removed? So the seismic moment is actually greater than the geodetic moment? Also, shouldn't the comparison be with the geodetic moment at the time of the Mw 7.1 earthquake instead of at the end of the 12 hours ?

Indeed, the estimated geodetic moment over the whole fault is 8.3E19 Nm while the estimated seismic moment over the whole fault is 9.5E19 Nm. We have changed the text to provide a better explanation for that discrepancy (line 203-218 of the new manuscript). In particular, we find that most of the seismic moment is released in the southern patch (~9.0E19 Nm). In that same region the geodetic moment is even lower (~4.5E19 Nm). We provide several explanation for that difference. First, seismic moment of the earthquakes in the GCMT catalog are obtained using PREM for the Earth structure, a model that differs from ours and that can over-estimate the rigidity especially at shallow depths (Bilek and Lay , 1999). Then, the Kalman filter used to process the GNSS has been tuned to properly recover slow processes such as afterslip. Therefore, it might not be suited to recover static offsets from large earthquakes, which can distort the recovery of the real ground motion in that case (Choi, 2007). Finally, as pointed out by Konca et al. (2007), there is also a moment-dip trade-off when using near-field geodetic data. Thus, the fact that our fault plane only approximates the real geometry of the Slab1.0 model could also explain the discrepancy. We have added this discussion in the main text (lein 210-231 in the revised manuscript).

- Figure 2: It would be useful to show the "time since mainshock" as a second x-axis. Please also indicate the moment magnitude of the two large aftershocks next to their dashed blue lines.

We do not think that it is relevant to show a second x-axis with « time since the mainshock » as the orange dots are by construction n-hours after the mainshock. However, we have added the moment magnitude of the two large aftershocks next to the dashed lines on the plots.

- Figures 3 & 5: the repetitive degrees around every subplots are redundant. Consider showing only the left and bottom ones. Consider better showing the trench with the usual chevrons. Indicate the meaning of the blue area in the legend. Add the moment magnitude of the aftershock next to the respective star.

Done

- Figure 4: Remove the title "postseismic 12 hours" as it shows afterslip distribution for longer periods.

Done

- Figure 7: This should be replaced by a composite with Figures S7.1 and S7.2. The corresponding discussion of the number of aftershocks and the logarithm of the same in the main text is not particularly useful. Instead, focus on the obvious difference between Figures S7.1 and S7.2.

We have taken that into account especially since this was also pointed out by all of the other reviewers. Figure 6 in the revised manuscript shows the evolution of the geodetic moment of afterslip along with the seismic moment released by aftershocks. The curves are on the same sub-figures by are displayed using distincts y-axis. That way, they look like normalized so that we can compare the temporal evolution, but the distinct y-scales allow the reader to get the actual value of the moment. We have kept a comparison between the geodetic moment of afterslip and the time evolution of the number of aftershocks (Figure 8) as we still discuss that relationship in the main text.

- Finally, please consider commenting the phenomenology shown in Figure S7.1. Why is the cumulative moment of aftershocks increasing so much in the northern segment around 6-7 hours? How does that translate in terms of fault slip? It does not seem clear from the various figures. It is hard to tell if the moment is significant because the plots use "normalized" time dependence. Since the geodetic and afterslip moment are so similar to the south, why not using moment (Nm) as the y-axis?

We think that this comment comes from the fact that the curves were normalized giving the false impression of a very large increase of seismic moment. We believe that with Figure 6 in the new manuscript, the fact that this increase is in fact very small should be more obvious.

**Reviewer Comment #1 by Bernd Schurr:**

- l.58 "seismic noise" – I think this is not really seismic noise but overwhelming signal, i.e. numerous often simultaneous aftershocks, that is causing problems to most detectors.

We have changed that part to better highlight the issues of detecting aftershocks right after a large mainshock (line 52-54 of the revised manuscript).

- l.28: change "activity" to "deformation". I think there is a "itself" missing after "express".

Done

- l.49: change "highly" to "more"

Done

- l.89ff: "The cumulative surface displacements are calculated at every hour since the mainshock origin time by computing the average positions over a 1-hour time window centered on the time of interest." Does cumulative refer here for cumulative during the one hour processed or cumulative since the mainsock. I assume the earlier, but please clarify.

We agree that the terminology that we use here is rather confusing. A given position in time on the position time series represents how much surface displacement has occurred since the mainshock. This is what we meant by « cumulative ». Therefore, the average position that we compute and use during the inversions represents how much surface displacement has occurred after N hour(s). Consequently, each snapshot on the former Figure 3 and 6 represent the total amount of afterslip that has occur after N hour(s). We have attempted to clarify that point in the text (line 100-104 in the revised manuscript).

- l.126: "yellow circles" should be "purple stars" ?

Done

- l.129: There is a word missing after "second". Maybe "patch".

Done

- The general fuzziness of both co-seismic and post-seismic slip models makes this assertion difficult to maintain (and the authors actually relativize it later in the paragraph). In particular, different fault model used (simple plain slab like the authors or varying slab dip based on e.g. slab2.0) in the modeling will shift location of slip. To start interpreting this, at least the modeling set-up of co- and post-seismic slip should be the same.

We do not think that it would be relevant to the study to add our own co-seismic slip model using our own dataset. There are already plenty of co-seismic models from other groups, that we show in Supplementary Materials S6, and which already illustrates the variability of the co-seismic slip area. To the first order, our model will very likely match the ones displayed and thus it will not change the discussion. Also, as we mention in the text, we are not making any assertion regarding the penetration of afterslip inside the co-seismic area by concluding that this is an observation for which we cannot reliably address the veracity (line 180 in the revised manuscript).

- First of all, slip in Fig. 3 and 6 I assume is the slip during the respective hour, not the cumulative slip added up also from the previous hours (must be based on the amplitudes and the fact that some patches vanish).

We think that our answer to comment l.89ff should clarify that point.

- I don't understand why, if only the offsets of the 2 largest aftershocks are corrected (hour 1 and hour 5), all slip vanishes in the southern patch also during the other hours. Please explain.

This relates to one of the point you have raised previously (l.89ff). The surface observations record the total amount of afterslip after n-hours. As we explain in the text, the observed afterslip to the south is for the most part due to the two largest aftershocks. So, by removing the signal from these earthquakes, there is consequently no slip in this region anymore.

- Fig. S7.1: The second M6.8 aftershock occurs during hour 5 and clearly shows up as a step in the graphed moment. However a step in slip seems to occur mainly in hour 6? Is this an averaging effect?

This is indeed an averaging effect. The second aftershock occurs at the end of the time window used for averaging the position at hour 5. Therefore, within that window, the majority of the data used for the averaging are not affected by the offset from that aftershock. Instead, the windows for hour 6 and after are fully offset by the aftershock. This explains the shift between the seismic moment and the geodetic moment. Also, instead of showing the continuous evolution of seismic moment, we were showing the seismic moment summed over time windows of 1 hour, to match the time window covered by the afterslip. This accentuated the effect even more. Thus, we have changed that figure (see Figure 6 in the revised manuscript).

- l.197ff: Please elaborate in one or two sentences what this predicted accelation phase signifies.

We have added a sentence to clarify what is this acceleration phase (line 256-257 in the revised manuscript).

- l.201ff: Please mention where the cited studies were based (Ecuador and Japan).

Done

- l.209: Change "rate-and-state law" to "rate-and-state friction law" here and everywhere else.

Done

- l.257: I wonder, are aftershocks anywhere actually operationally forecasted based on some models (maybe a citation would be good)? If so, I would assume that models have to be simple and robust. Here e.g. CFS would naturally predict aftershocks around the co-seismic rupture area, where they do occur, for the Illapel eqk and also for many other subduction zone earthquakes. I wonder, how realistic and it is to actually do the hindsight analysis outlined here in near real time and if it really adds value. Of course, this could be tested.

It is indeed, very difficult to predict the added value of including information about very early afterslip for forecasting aftershocks, and the point that we want to make is that we have now the capability to investigate such question. However, it is true that we might have been too ambitious by having the word « operational ». In fact, we are not aware if this is done anywhere. We have rephrased that last sentence to reflect better our point (line 319-321 of the revised manuscript).

- l.271ff: "Our additional finding is that the slip patterns that we observe after 12 hours persists over the first 2 months. When that is the case, information about very-early post-seismic slip can help to characterize longer-lasting post-seismic slip, which can prove to be useful to include for the forecast of aftershocks locations." But can this really be generalized?

A community reviewer pointed out to us that the fact that afterslip patterns seem rather stationary in time, and that is the case for many examples. Following that question, we have investigated the literature on the question. We have only found one study that suggest a hint of afterslip migration (see line 157-161 of the revised manuscript). Therefore, although we cannot conclude with certainty that this can be generalized, it seems reasonable to make the assumption that afterslip is commonly stationary. We have added a sentence to reflect that (line 340-342 of the revised manuscript).

**Community Comment #2 by Dietrich Lange:**

- In the manuscript, you sometimes refer to "south/north of the rupture area," but it seems that the two patches you classify as south and north of the rupture are still located in the co-seismic region (e.g., between 30-32°S). I suggest clarifying the two patches exact position and how they are related to the co-seismic slip.

We have highlighted the regions that we are referring to on Figure 3 so that the reader can clearly see the location of what we refer to as the southern patch and the northern patch. We have also added that added information in the text (line 143 and line 149 in the revised manuscript).

- In particular, previous authors observed a widening of the aftershock zone of the Illapel earthquake (e.g., Lange et al., 2016, GJI, doi: 10.1093/gji/ggw218). It would be interesting to state if this is observed with the catalog based on template matching and how this relates to the inverted afterslip. I assume that the possible expansion outside the co-seismic region is too small to be robustly inverted by the hourly evolving afterslip model. I suggest adding some words to this.

This is an interesting point that you raise regarding the migration of aftershocks. Huang et al. (2017) as well as Frank et al. (2017), from which the catalogs of aftershocks used in our studies are taken, seem to make a similar observation, even when looking at the very early stage. However, regarding the afterslip, we do not see such migration. Instead, as mentioned in the text, we only observe a growth of the afterslip patches (see line 142 in the revised manuscript). It is likely that our afterslip maps don't allow to investigate such fine details. Because we already mention in the text that we do not observe any afterslip migration, we have decided not to discuss that point in the main text.

- In particular, the authors find that the partitioning in seismic and aseismic slip changes in time (As mentioned in Line 207). I could not feasible see this in Figure 7 and S7.1, and suggest to simply plot the displacement versus the number of events for the patches to show the relationship between both processes. For example, Lange et al. (2014, GJI, doi: 10.1093/ gji/ggu292) mapped for the late postseismic (e.g. >1d) of the Maule 2010 earthquake the partitioning of seismic to aseismic slip, which was relatively stable in time.

We would like to clarify that we do not argue that the seismic/aseismic slip partitioning changes over time. Line 207 in the former manuscript only relates to the mechanical link that has been proposed between afterslip and aftershocks. One usual argument for the fact that afterslip is mechanically driving aftershocks is based on the fact that the shape of the time evolution of afterslip closely match the shape of the time evolution of the cumulative number of aftershocks. But, this is not what we observe. Figure 7 (now Figure 8 in the revised manuscript) shows that : the afterslip

(blue line) evolves clearly differently than the number of aftershocks (orange dashed line).

- There is a similar observation for strike-slip faulting for early observation of afterslip and aftershocks, and their relation (Savage, 2007, GRL, doi: 10.1029/2010GL042872) shows that aftershock seismicity rate is not proportional to the stress relaxation rate for the San-Andreas fault.

Thank you for pointing that study to our attention. At the time of writing the first version, we have attempted to find other cases exhibiting such behavior, without success. Thus, we have added that to the main text (line 268-270 in the revised manuscript).

- I suggest adding a caption to Fig. S7.1 and S7.2. Figure S7, S7.1, and S7.2 (right panels) might need additional labeling for the number of aftershocks. Currently, only the slip is labeled.

This is done on Figures 6, 8 and S7.2 in the revised manuscript and Supplementary Materials.

- Table 1 and 2 do not contribute and might belong to the supplementary.

Done (see Supplementary Material S1).

- I suggest showing Figure S7.1 (temporal development southern patch) in the primary material since parts of the findings are difficult to understand without this figure.

Done (see Figure 6 in the revised manuscript).

- Line 9: …. and thus could contribute to a more data-drivenforecasts of long-term aftershocks. I cannot necessarily follow the argument in Line 9. Since the partitioning of seismic slip in some places changes in time prediction might be very difficult if this process remains enigmatic. Does the suggested forecast suggested here use Omori-laws and constant b-values, such as modelled by (Jonsdottir et al., 2006, Tectonophysics 424, https://doi.org/10.1016/j.tecto.2006.03.036.)?

We would like to clarify again that we do not argue that the seismic/aseismic slip partitioning changes over time. However, we do agree that this last sentence of the abstract might be too ambitious of an opening with respect to our finding. We have rephrased that last sentence (line 9-11 of the new manuscript).

**Reviewer Comment #2 by Mathilde Radiguet:**

- It would be useful to add on Figure 3 the coseismic slip distribution estimated from previous studies (for example the one inferred by Melgar et al. 2016.

Done as the shaded blue region in Figure 3

- Of course uncertainties exist on this slip distribution based on the data, inversion scheme and fault geometry used by previous authors, but the authors could show several models if needed.

Done in the Supplementary Material S6

- The authors could also perform their own inversion of the co-seismic slip (using only GNSS data). Even if this inversion would be constrained only by geodetic data, it would be interesting because fully consistent with the post-seismic study in terms of fault geometry, Green's function and data with the post-seismic study. it could be added to the supplementary material.

We do not think that it would be relevant to the study to add our own co-seismic slip model based on the data that we have. There are already plenty of co-seismic models from other groups, that we show in Supplementary Material S6, and which already illustrates the variability of the region of co-seismic slip. To the first order, our model will very likely match the ones displayed and thus it will not change anything to the discussion.

- l. 104: "We search for the spatial distribution of slip amplitude and rake angle independently for each time step": the slip amplitudes obtained are shown but not the rake angles. Do they vary from one time step to another ? The optimal rake for each time step should be given in the supplementary materials.

We have included that information in Supplementary Material S4

- You do not want to add that you invert the rake in a +/- 15° of the convergence direction in the main text or the legend of the figure S4.17

We have included that information in the main text (line 119-121 in the revised manuscript).

- l.181 "a seismic moment of 9.5x 10^19 Nm." what is the region considered for the calculation ? Is it the same as the one shown in Fig. S6.2 ?

We have done a better description of the area considered to obtain this estimate (line 208-214 in the revised manuscript)

- l. 189-190: same question, what is the area considered ?

We have highlighted the regions that we are referring to on Figure 3 in the revised manuscript so that the reader can clearly see the location of what we refer as the southern patch and the northern patch. We have also added that into the text (see line 143 and line 149 in the revised manuscript).

- In Fig. S7, the cumulative slip could be converted to seismic moment so that one can see where the values given in the text for the seismic/aseismic rations come from. The time series from Fig.7 could also be included in the main text, as they are really relevant for the discussion of the paper.

We have made that change by including a new figure in the revised manuscript (Figure 6).

- On FigureS7.2: what is happening between 6h and 7h  (strong increase in cumulative moment): is there a large aftershock at this time ?

We think that this comment comes from the fact that the curves were normalized giving the false impression of a very large increase of seismic moment. With the new figure 6, we believe that it clarifies that the increase is in fact very small.

- "yellow circles" should be "pink stars »

Done

- Several problems with figure/table numbers (they appear with "??"): Line 85, 136, 165…

Fixed

---

## Author Response (AR2)

Dear editor and reviewers,

We would like to thank you for providing additional comments about our manuscript se-2021-6 untitled « Very early identification of a bimodal frictional behavior during the post-seismic phase of the 2015 Mw8.3 Illapel, Chile, earthquake ». Below, you will find a point-by-point response to all the comments. The questions/comments are in blue while our responses are written in black. Note that the line numbers are taken from the manuscript with tracked-changes.

Referee #1 : Bernd Schurr

— — —

Referee #3 : Sylvain Barbot

[0] Some important reviewer suggestions, such as constructing their own coseismic slip distribution, were ignored.

Although we argued in our previous revision that we do not see the relevance of producing our own co-seismic model, we have decided to add to the Supplementary Materials our own co-seismic model (see changes in Supplementary Materials S6) since this has been pointed out by several reviewers. We have also added that information in the main text (line 182 in the revised manuscript). We also include the co-seismic static offsets used to obtain the model (see Table S3 in Supplementary Materials S1). The conclusions remain unchanged.

[1] Line 16: A recent contribution to this debate includes

Ozawa, S. and Ando, R., 2021. Mainshock and Aftershock Sequence Simulation in Geometrically Complex Fault Zones. Journal of Geophysical Research: Solid Earth, 126(2), p.e2020JB020865.

but other work can be discussed in more detail.

[2] Line 17: This sentence is vague and the discussion could be more thorough, including more recent contributions. For example, Bath law describes the maximum size of aftershocks, although it does not always work.

[3] Line 19: "All of that" is too colloquial. The few introductory paragraphs could be better phrased to highlight current knowledge on the physics of aftershocks.

Based on these comments, we have attempted to re-write the first two paragraphs of the manuscript so that it reflects better the current knowledge regarding aftershocks behavior (line 13-46).

[4] Line 114: It is still unclear why a MCMC approach is used when the problem is entirely linear and therefore has a closed-form solution. See the work of

Hang, Y., Barbot, S., Dauwels, J., Wang, T., Nanjundiah, P. and Qiu, Q., 2020. Outlier-insensitive Bayesian inference for linear inverse problems (OutIBI) with applications to space geodetic data. Geophysical Journal International, 221(1), pp.334-350.

for linear problems that need nonlinear optimization.

We agree with the comment that when the problem is entirely linear with Gaussian uncertainty on the data and on the prior model, there is a closed-form solution. However, there are two reasons why it is not applicable in our case : (1) we assume a uniform prior on the model and (2) we use a positivity constraint on the slip, which truncate our prior distribution. This is why we believe that a MCMC approach is more suited for our problem. We have added some clarification in the main text (line 130-132). As pointed out, Hang et al. (2020) provides a way to overcome the issues mentioned above, but we were not aware of that study at the time of submission. Regarding this study, we do not think it is worth re-doing all the work using that approach,

especially considering the potential time necessary to get use to the suite of programs. But, we will investigate how it can potentially be used for our future works.

This is an issue from the plotting routine that we have now fixed on all the Figures in the main text and the Supplementary Materials.

Done on Figure 6 and all other relevant Figures.

[7] Although I cannot make any more informed suggestions on the matter, is there anything in the offshore or ashore geomorphology that indicates structural controls of coseismic and afterslip dynamics? For example, this section of the Chile subduction zone is an erosive margin, implying a rough plate interface.

We agree with the comment that it is difficult the make strong hypothesis regarding what could control the behavior of afterslip and coseismic slip. We already attempted to provide some hypothesis in the main text (line 269-283). For instance, we highlight the study of Poli et al. (2017), that shows evidence for strong fracturation in this region. The way fluids circulate in this region could have played a role at controlling coseismic slip and afterslip. We also point out that coupling is different for each of the afterslip patch (Métois et al., 2014). This could also maybe explain the bimodal behavior that we obtain. Regarding the roughness of the plate interface, it is true that this could also be proposed to explain the behavior of coseismic slip and afterslip (although we can only make hypothesis from our study). We have added that in the main text citing the study of Comte et al. (2019), which provides evidence that materials are entering the subduction interface at the region of Illapel. We also point to the study by Lange et al. (2016) who also suggests that roughness of the plate interface might play a role at controlling the spatial pattern of long-term aftershocks following the Illapel earthquake (line 274-279).